Resource

# LuminoCell: a versatile and affordable platform for real-time monitoring of luciferase-based reporters

Kamila Weissová[1], Bohumil Fafílek[2,3,4], Tomasz Radaszkiewicz[5], Canan Celiker[1], Petra Macháčková[6], Tamara Čechová[3,4], Jana Šebestíková[1,5], Aleš Hampl[1,4], Vítězslav Bryja[5], Pavel Krejčí[2,3,4], Tomáš Bárta[1,5]

Luciferase reporter assays represent a simple and sensitive experimental system in cell and molecular biology to study multiple biological processes. However, the application of these assays is often limited by the costs of conventional luminometer instruments and the versatility of their use in different experimental conditions. Therefore, we aimed to develop a small, affordable luminometer allowing continuous measurement of luciferase activity, designed for inclusion into various kinds of tissue culture incubators. Here, we introduce LuminoCell—an open-source platform for the construction of an affordable, sensitive, and portable luminometer capable of real-time monitoring in-cell luciferase activity. The LuminoCell costs $40, requires less than 1 h to assemble, and it is capable of performing real-time sensitive detection of both magnitude and duration of the activity of major signalling pathways in cell cultures, including receptor tyrosine kinases (EGF and FGF), WNT/$\beta$-catenin, and NF-$\kappa$B. In addition, we show that the LuminoCell is suitable to be used in cytotoxicity assays as well as for monitoring periodic circadian gene expression.

## Introduction

Luciferase reporter assays allow the study of a wide range of biological processes in cells and tissues. These reporter systems use luciferins, a class of small molecules that react with oxygen in the presence of the enzyme luciferase, to release the energy in the form of light. Luciferase assays represent a well-established experimental approach in cell and molecular biology, commonly used for gene expression analyses, promoter analyses, cytotoxicity assays, and signal transduction analyses. These assays are sensitive, reproducible, compatible with a range of internal control reporters, and provide a broad linear dynamic signal range (Smale, 2010). The extent of the application of luciferase assays in research is demonstrated by the number of publications that reference the luciferase reporters; more than 47,000 research articles were found at Pubmed (keyword: "luciferase reporter," December 2021), with more than 31,000 articles published in the past 10 yr.

In luciferase reporter assays, the luciferase activity is determined by a luminometer device, usually in a bench microplate or a single tube reader configuration. However, high costs of commercial luminometers may limit their availability for many research groups. In addition, there are other limitations associated with these devices: (I) the end-point measurement is often the only possible way of analysis and it does not provide information about the temporal dynamics of the signal; (II) most instruments are difficult to sterilize and cannot be placed in the cell incubator, and thus are not suitable for long-term experiments with growing cell cultures; (III) the measurement often requires cell harvest and lysis; (IV) complex experimental schemes, such as multiple treatments at different times, are difficult to perform with the standard trap-door luminometer devices.

Here, we introduce an open-source platform for construction of a versatile, cheap, light-weight, and portable luminometer—the LuminoCell—that can be 3D printed and assembled within 1 h. The LuminoCell enables continuous in-cell monitoring of luciferase activity in the growing cell cultures, allowing the study of dynamic biological processes in real time. We also provide proof-of-concept experiments of the device use, in monitoring the activity of cell signalling pathways, cytotoxicity assays, and gene expression assays.

## Results

### The LuminoCell description

The central part of the LuminoCell is represented by the light-to-frequency converter TSL237S-LF (Mouser) that is positioned in the centre of 3D printed wells in a plastic case (Fig 1A–C). This type of sensor has already been reported for light measurement

[1]Department of Histology and Embryology, Faculty of Medicine, Masaryk University, Brno, Czech Republic   [2]Institute of Animal Physiology and Genetics of the Czech Academy of Sciences, Brno, Czech Republic   [3]Department of Biology, Faculty of Medicine, Masaryk University, Brno, Czech Republic   [4]International Clinical Research Center, St. Anne's University Hospital, Brno, Czech Republic   [5]Department of Experimental Biology, Faculty of Science, Masaryk University, Brno, Czech Republic   [6]Cellular Imaging Core Facility, Central European Institute of Technology (CEITEC), Masaryk University, Brno, Czech Republic

Correspondence: tbarta@med.muni.cz

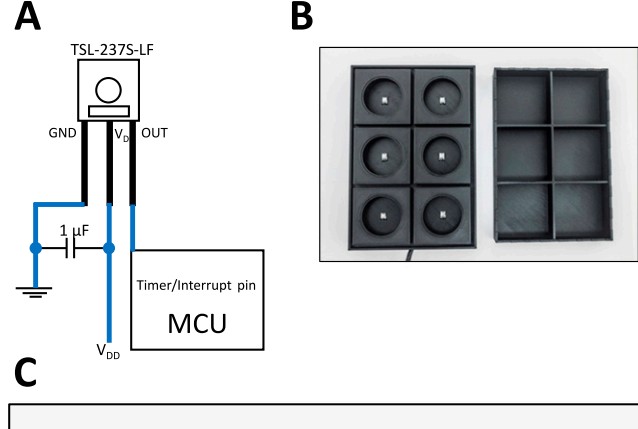

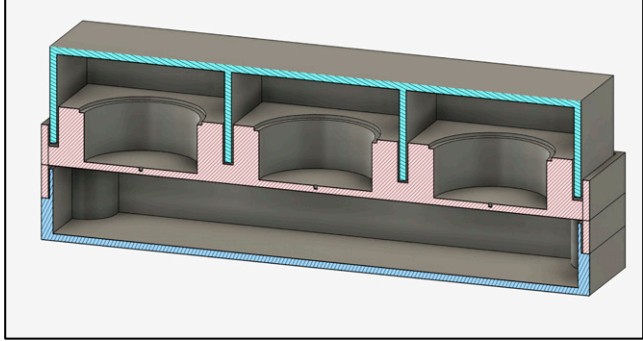

**Figure 1. The LuminoCell description.**
**(A)** General wiring diagram. **(B)** The LuminoCell with lid (top view). **(C)** Cross section of the LuminoCell. green–lid, red–middle part, blue–bottom part.

application in biology (Droujko & Molnar, 2022). A light-to-frequency converter like the TSL237S-LF, converts light intensity into a series of square-wave pulses, with the frequency depending on the light intensity. The limit of light sensitivity is determined by the on-sensor noise, resulting in occasional spurious pulses even without incoming light (dark frequency, $f_D$). The typical $f_D$ of TSL237S-LF, according to the manufacturer, is 0.1 Hz and the maximum operating frequency is 1 MHz, providing a dynamic range of seven orders of magnitude per 1 s of the measurement process. Pulses, generated by TSL237S-LF upon light detection, are integrated by the Arduino microcontroller unit (MCU) (www.arduino.cc) that is positioned inside the LuminoCell. The luminometer is placed into a cell incubator and it is connected to a computer outside an incubator using a thin USB cable. The cell culture Petri dishes, containing luciferase reporter cell lines, are positioned on the device and covered by a lid to protect potential incoming light, while allowing gas exchange (Fig 1B and C). The device can be reused with a new cell samples multiple times just by changing petri dishes. In addition, one can perform treatment of cell cultures and continue with a measurement allowing to study dynamic changes to the luciferase activity in real time. Luciferase activity (number of pulses) is monitored using a serial monitor provided by Arduino IDE software (www.arduino.cc). The LuminoCell is capable of simultaneously measuring the luciferase signal in six 40 mm petri dishes and estimated costs for assembling the LuminoCell are ~40 USD (the costs for the 3D printed case are not included) (Table S1). The general wiring diagram of

the LuminoCell is shown in Fig 1A (details are depicted in Figs S1 and S2).

## Proof-of-concept experiments

We aimed to test the performance of the LuminoCell using a wide spectrum of different luciferase reporters and cell lines. We tested the LuminoCell in monitoring the activity of fibroblast growth factor receptor (FGFR) signalling in cells, using the pKrox24[Luc] luciferase reporter. The pKrox24 contains promoter sequences based on the *EGR1* upstream of the firefly luciferase, and was developed to record the FGF-mediated activation of the RAS-ERK MAP kinase pathway (Gudernova et al, 2017). Rat chondosarcoma (RCS) cells, stably expressing the pKrox24[Luc] (RCS::pKrox24[Luc]), were used to assess the signal background, defined as the number of pulses (dark frequency, $f_D$), in the absence of luciferin. Six to seven pulses during a 5-min integration time were recorded, which corresponds to $f_D$ = ~0.02 Hz (Fig 2A). When luciferin was added to the culture medium, the number of recorded pulses increased to 15–18/5 min ($f_D$ = ~0.06 Hz) reflecting the basal pKrox24[Luc] transactivation. Addition of the recombinant FGFR ligand FGF2 led to a profound increase in the number of detected pulses, reaching its maximum at 142 pulses/ 5 min after 5 h (300 min) of FGF2 addition (Fig 2A). To remove the background, caused by the basal EGR1 activity in the absence of FGF2, we refined the data and subtracted the background noise (normalized to the number of pulses before the addition of FGF2) (Fig 2B). All the other data sets, shown in this article, are presented with subtracted backgrounds. This experiment clearly demonstrates that the LuminoCell is capable not only to monitor kinetic changes of the luciferase activity in real-time, but also allows researchers to pick the right time for potential downstream analyses.

We evaluated different cell densities to test the sensitivity of signal detection. The RCS::pKrox24[Luc] cells were seeded at concentrations ranging from $3 \times 10^3$ to $1 \times 10^5$ cells/cm². The cells were stimulated with FGF2 and luciferase activity measurement started immediately upon FGF2 addition. The FGF2-mediated increase in signal was recorded even at the lowest used cell density ($3 \times 10^3$ cells/cm²), generating 8 pulses/5 min above the background (Fig 2C). Extending the pulse integration time from 5 to 20 min led to an approximate fivefold increase in luminometer sensitivity (8 versus 30 pulses at $3 \times 10^3$ cells/cm²), with a negligible effect on temporal resolution of the signal (Fig 2D). Thus, the integration time may be varied to increase luminometer sensitivity, to detect the signal of weak promoters, dim luciferase reporters or a low number of cells.

To demonstrate the LuminoCell capacity to monitor dynamic changes of the reporter activity, we treated RCS::pKrox24[Luc] cells with ARQ087, a potent inhibitor of FGFR catalytic activity (Balek et al, 2017). ARQ087 was added into the culture media 30 min before FGF2, with luciferase measurement startingimmediately upon the FGF2 addition (Fig 2E). We found a substantial inhibition of the pKrox24[Luc] transactivation in cells treated with ARQ087. This was confirmed by Western blot analysis of FGF2-mediated ERK MAP kinase phosphorylation in RCS::pKrox24[Luc] cells, which was inhibited by ARQ087 (Figs 2F and S3).

The pKrox24[Luc] responds to activation of the RAS-ERK module and therefore can be transactivated by many cell signalling pathways which use ERK, including as many as 30 different receptor

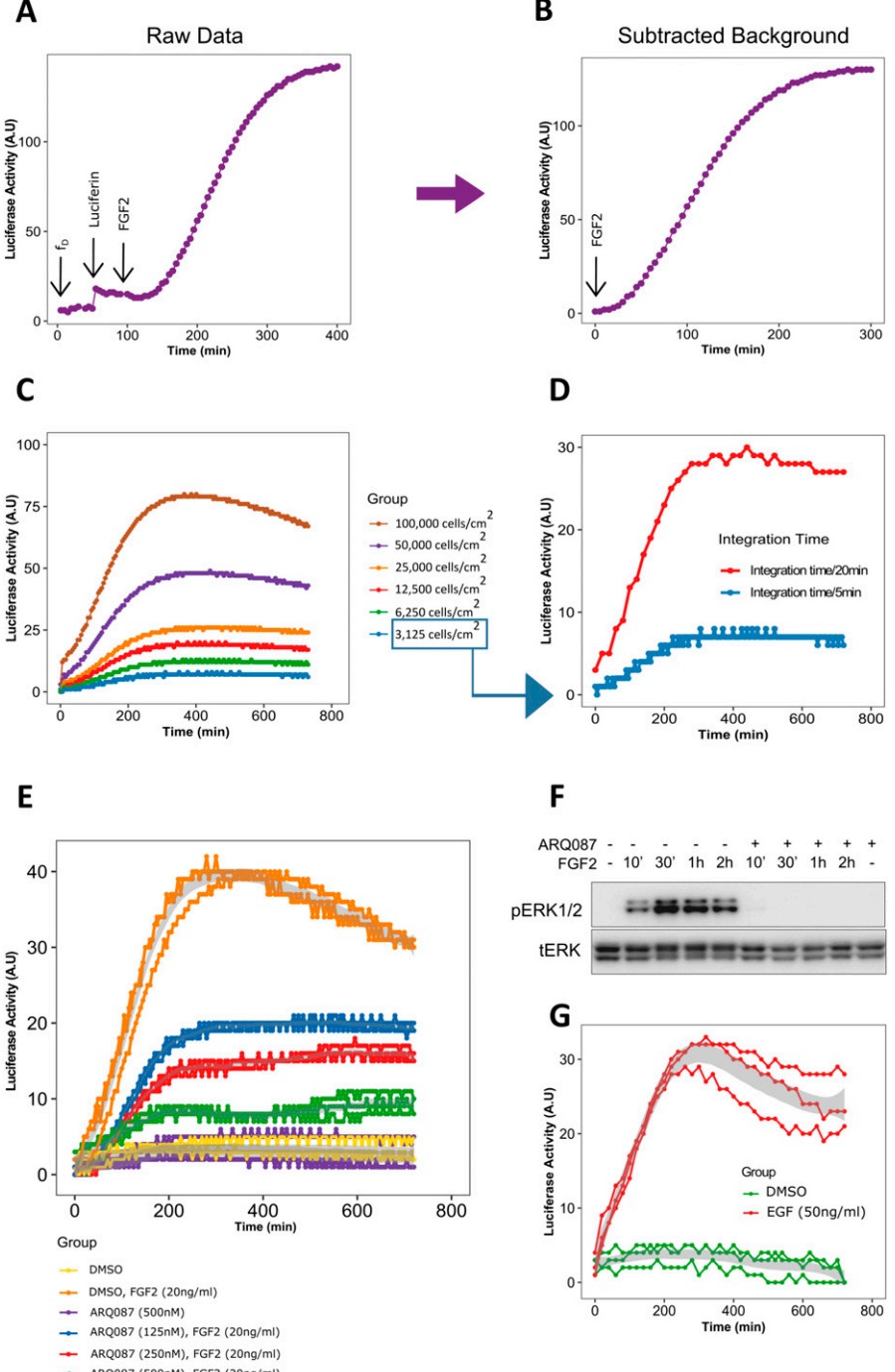

**Figure 2. Transactivation of pKrox24^Luc by fibroblast growth factor receptor and epidermal growth factor receptor signalling.**

**(A)** Luciferase activity upon addition of luciferin and recombinant FGF2 (arrows) to RCS::pKrox24^Luc cells. $f_D$, background (dark) frequency. Integration time for pulse count was 5 min, if not stated otherwise. **(A, B)** Normalized data from (A), with subtracted background. **(C)** FGF2-mediated pKrox24^Luc transactivation in different cell densities. **(D)** 3,125 RCS::pKrox24^Luc cells/cm² were stimulated with 20 ng/ml FGF2, and the luciferase activity was measured using a 5 or 20 min integration time. **(E)** Luciferase activity in the presence of 20 ng/ml FGF2 and different concentrations of fibroblast growth factor receptor signalling inhibitor ARQ087 in RCS::pKrox24^Luc cells. **(F)** Western blot analysis of ERK phosphorylation (p) in RCS::pKrox24^Luc cells treated with 500 nM ARQ087 and 20 ng/ml FGF2. Total levels of ERK (t) serve as a loading control. Uncropped images are shown in Fig S3. **(G)** pKrox24^Luc transactivation mediated by EGF in 293T::pKrox24^Luc cells. Integration time 10 min. Data from three replicates are shown.

tyrosine kinases (Gudernova et al, 2017). We tested whether the luminometer is able to monitor the activity of a receptor tyrosine kinase unrelated to FGFR, the epidermal growth factor receptor. The 293T::pKrox24^Luc cells were cultured in the presence of EGF and luciferase activity was monitored. Fig 2G shows the pKrox24^Luc transactivation mediated by epidermal growth factor receptor signalling.

We tested the activity of the canonical WNT/β-catenin pathway, using Super-Top-Flash (STF) cells stably expressing luciferase reporter under the control of seven LEF/TCF–binding sites (Xu et al,

2004). Cells were cultured in the presence or absence of GSK3α/β inhibitor CHIR99021, a potent activator of the canonical Wnt-signalling cascade (Bennett et al, 2002). We detected an increase in luciferase signal 3 h (180 min) upon CHIR99021 addition (Fig 3A). The signal increased to 23–29 pulses after 10 h (600 min), followed by a decrease to 10–15 pulses at the end of experiment, 16.5 h (1,000 min), whereas the control increased from four pulses at the start of measurement to 4–8 pulses at the end of experiment. End-point analysis of the Wnt signalling luciferase reporter activity using a

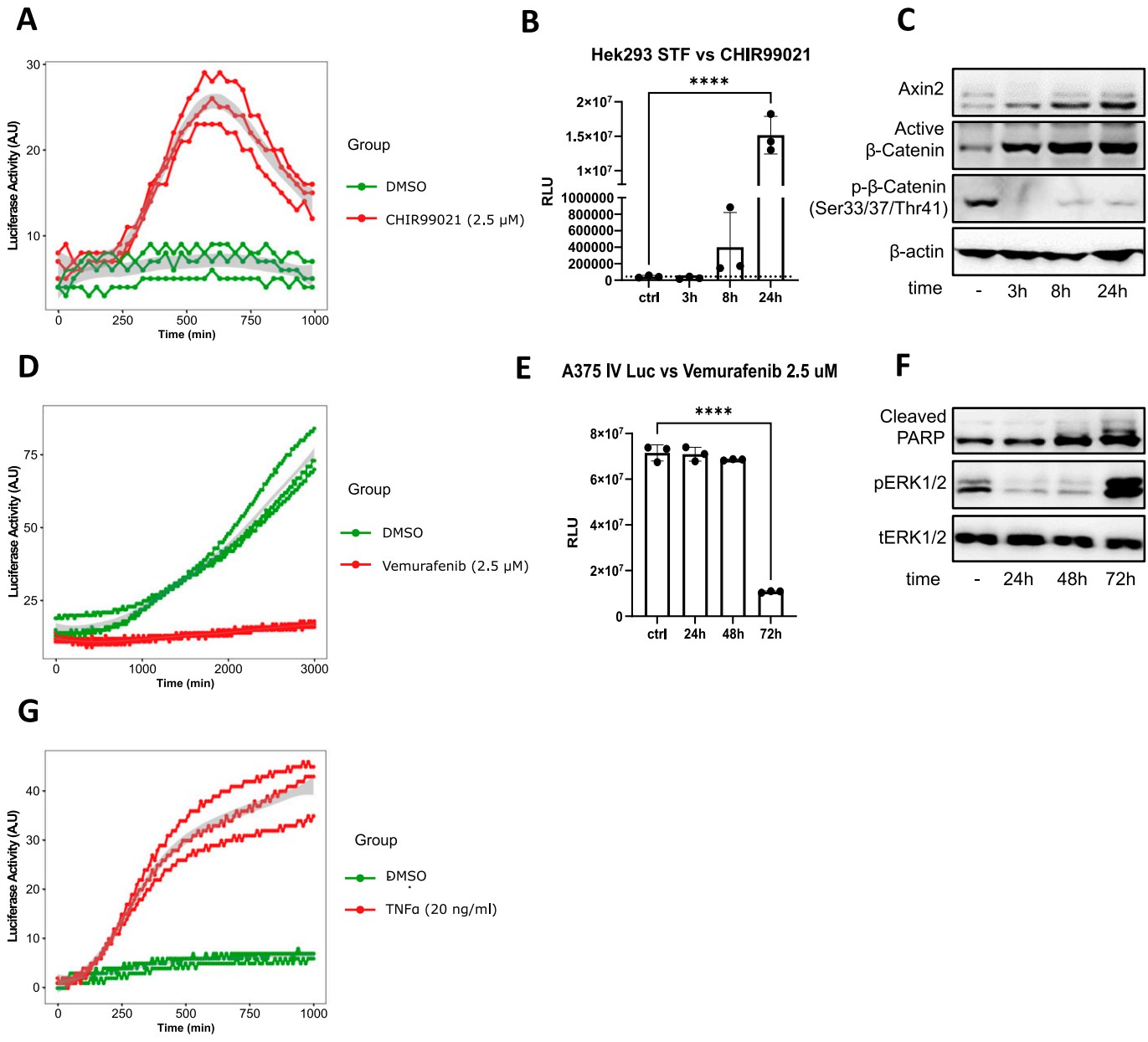

**Figure 3. Testing the ability of the LuminoCell to measure luciferase activity of Wnt, TNFα pathways reporters, and for cytotoxicity testing.**
**(A)** Assessment of canonical Wnt signalling activity in Super-Top-Flash (STF) cells in the presence of CHIR99021 or DMSO as a control. Luciferase activity measured using the LuminoCell (integration time 30 min). Data from three replicates are shown. **(B)** End-point analysis of canonical Wnt signalling activity in STF cells in the presence of CHIR99021 or DMSO as a control (ctrl). Luciferase activity measured using bench-top luminometer (Hidex Bioscan). **(C)** Western blot analysis of β-catenin and Axin2 expression in STF cells, β-actin represents a loading control. The data show representative blots of three independent experiments. Uncropped images are shown in Fig S4. **(D)** Cytotoxicity testing using A375-IV$^{Luc}$ cells. Cells were cultured in the presence of vemurafenib or DMSO as a control. Luciferase activity measured using the LuminoCell (integration time 20 min). Data from three replicates are shown. **(E)** End-point analysis of cytotoxicity using A375-IV$^{Luc}$ cells. Cells were cultured in the presence of vemurafenib or DMSO as a control. Luciferase activity measured using bench-top luminometer (Hidex Bioscan). **(F)** Western blot analysis of PARP cleavage and ERK (p) phosphorylation. Total (t) ERK represents a loading control. The data show representative blots of three independent experiments. Uncropped images are shown in Fig S5. **(G)** Assessment of NF-κB activity in the presence of TNFα using 293T::NF-κB$^{Luc}$ cells. Luciferase activity measured using the LuminoCell (integration time 10 min). Data from three replicates are shown.

bench-top luminometer confirmed activation of the Wnt signalling pathway 3 h after CHIR99021 treatment (Fig 3B). In addition, Western blot analysis confirmed the activation of the Wnt canonical pathway 3 h after CHIR99021 treatment, demonstrated by accumulation of β-catenin that is associated with a decrease of its phosphorylation and followed by stabilization of its target gene *AXIN2* (Figs 3C and S4).

Our experiments demonstrated the ability of the LuminoCell to monitor dynamic changes of the luciferase reporters for major signalling pathways. Next, we aimed to test the LuminoCell for its ability to be used for cell cytotoxicity assay. The melanoma cell line A375-IV$^{Luc}$ (Kucerova et al, 2014) continuously expressing luciferase vector pGL4.50 (luc2/CMV/Hygro) (Promega) was cultured in the

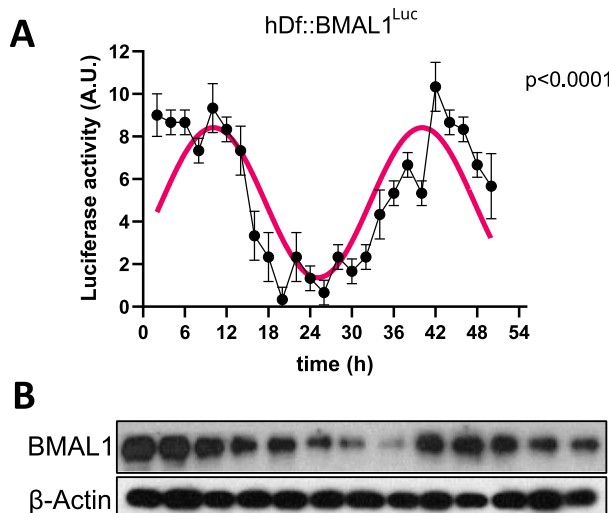

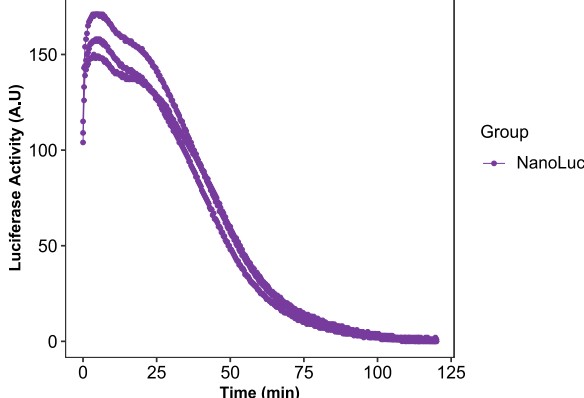

**Figure 5. Monitoring NanoLuc activity in ARPE-19 cells expressing NanoLuc (integration time 20 min).**
Data from three replicates are shown.

**Figure 4. Testing the ability of the LuminoCell to quantify the activity of circadian _BMAL1_ luciferase reporter.**
**(A)** Assessment of _BMAL1_ expression using luciferase reporter cell line hDf::BMAL1^Luc upon serum shock (integration time 2 h). Error bars show mean ± sd from three replicates. **(B)** Expression of BMAL1 upon serum shock, as demonstrated using Western blot. _β_-actin represents loading control. The data show representative blots of three independent experiments. Uncropped images are shown in Fig S6.

presence of vemurafenib—an inhibitor of the constitutively active B-Raf V600E proto-oncogene, used for the treatment of late-stage melanoma (Flaherty et al, 2010; Chapman et al, 2011). Vemurafenib blocks cell proliferation, induces cellular stress and senescence of melanoma cells via inhibition of the B-Raf/MEK/ERK pathway (Haferkamp et al, 2013; Peng et al, 2017; Su et al, 2020). We detected an increase of luciferase activity in the control condition (DMSO) after 11 h (660 min) from the start of the experiment that kept increasing to 87 pulses, indicating an increase of cell proliferation. However, in the presence of vemurafenib we detected low luciferase activity ranging from 14 to 19 pulses at the end of the experiment, indicating blocked cell proliferation (Fig 3D). End-point analysis using a bench-top luminometer revealed a decrease of luciferase activity 72 h upon vemurafenib treatment (Fig 3E). Western blot analysis revealed increased apoptosis associated with increased PARP cleavage 48 h after treatment and elevated phosphorylation of ERK (Su et al, 2020; Yue & López, 2020), indicating induction of cellular stress and paradoxical reactivation of the MAPK pathway in response to vemurafenib treatment (Su et al, 2020) (Figs 3F and S5).

We tested the activity of canonical NF-κB signalling in 293T cells stably expressing the NF-κB^Luc reporter (293T::NF-κB^Luc). Cells were treated by recombinant TNFα that represents a potent activator of NF-κB (Schütze et al, 1992). The increasing intensity of the luciferase signal was recorded for at least ~16.5 h (1,000 min), generating up to 46 pulses/10 min (Fig 3G). Cells in the absence of TNFα increased the luciferase signal from 0 to 2 pulses/10 min to 6–7 pulses/10 min at the end of the experiment at 16.5 h.

We tested the LuminoCell for the ability to record circadian oscillations in human dermal fibroblasts (hDf). The use of luciferase

reporters in circadian research highly simplifies and cuts the expenses for these long-term experiments, allowing the study of circadian rhythmic gene oscillation in vitro in real time. _BMAL1_ (_ARNTL_) is one of the key components of the circadian molecular clock and its expression exhibits a rhythmic pattern with an approximately daily period. We used luciferase _BMAL1_ expression reporter and generated the hDf::BMAL1^Luc reporter cell line using the lentiviral transduction approach. Confluent hDf::BMAL1^Luc cultures were synchronized using serum shock, a widely used approach to synchronize molecular circadian machinery (Balsalobre et al, 1998), and luciferase activity was monitored for 52 h. Using the LuminoCell, we were able to detect high-amplitude circadian oscillations of _BMAL1_ expression in hDFs cultures (Fig 4A). This amplitude of _BMAL1_ expression was confirmed using Western blot analysis (Figs 4B and S6).

NanoLuc luciferase reporter system offers several advantages over established luciferase systems, including enhanced stability, smaller size, and >150-fold increase in luminescence (England et al, 2016). To test the LuminoCell for the ability to detect signal from NanoLuc reporter, we used ARPE-19 cells expressing NanoLuc vector. Addition of the NanoLuc substrate led to 110 pulses/20 min, reaching the maximum of 170 pulses/20 min after 7 h of measurement (Fig 5). The signal gradually decreased presumably because of the depletion of the substrate.

## Discussion

Here we introduce an open-source platform for construction of an affordable and simple LuminoCell, a luminometer capable of real-time monitoring of luciferase activity in living cells. We provide a full description of the device, 3D print files, and the complete source code shared via public repository. In addition, we also provide a set of proof-of-concept experiments that test the LuminoCell functionality. Five different luciferase reporters expressing the firefly reporter gene _luc2_ and one reporter expressing NanoLuc were used to test the luminometer. Both luciferase systems are commonly used and have light emission centred at 560 and 460 nm, respectively (England et al, 2016). However, there are also other

systems that use luciferase genes derived from different species including *Pyrophorus plagiophthalamus*, *Renilla reniformis*, and *Gaussia princeps*. All these systems have the emission peak between 460 and 600 nm (Gil et al, 2012). Given the broad detection range of different wavelengths of light (320–1,050 nm), the LuminoCell is capable of detecting other light emission–based reporter systems; however, one should test if the given reporter system is applicable to be used with this luminometer.

The LuminoCell provides a sufficient theoretical dynamic range of seven orders of magnitude (0.1 Hz–1 MHz), and it is comparable with expensive commercial devices that possess a dynamic range of about eight orders of magnitude. Although we have not compared the LuminoCell with other commercial devices, it is of note, however, that this luminometer cannot compete with commercial instruments in terms of detection limits. Whereas this simple luminometer has a detection limit around 40 pW/s of light irradiance (TSL237S-LF has irradiance responsivity of 2.3 kHz/[$\mu$W/cm$^2$]), devices equipped with expensive PMT tubes, or special CCD cameras offer much better sensitivity in attowatts (Enomoto et al, 2018). The relatively low detection limit of light irradiance is, however, estimated for 1 s of integration time. Therefore, one can improve the detection limits and dynamic range of this simple luminometer by increasing the integration time that is necessary to count pulses generated by TSL237S-LF, as described in Figs 2D and 4A. However, this approach will lead to a decrease of temporal resolution, which is still sufficient to monitor dynamic changes of the luciferase activity during experiments.

Among the most beneficial advantages of the LuminoCell include: its price, versatility, small size, potential customization, and expandability. Whereas the prices for commercial devices range from thousands to tens of thousands of USD, this luminometer provides a cheap and affordable solution for any research group. In addition, commercial devices very often do not allow simultaneous real-time measurements in living cells and require end-point analysis often associated with cell lysis. The LuminoCell allows real-time measurements that do not require cell lysis, providing to a researcher the possibility to perform treatment of the reporter cell line during measurements and to continue to monitor dynamic changes of the luciferase activity upon treatment. In addition, due to its compact size, the LuminoCell can be positioned into various kinds of tissue incubators including CO$_2$ cell incubators, multi-gas incubators, hypoxia stations, and other specialized culture systems. Because of its portability, the LuminoCell can be used in a wide range of scientific disciplines ranging from cell and molecular biology to ecology for monitoring pollutants in the environment. Importantly, commercially available luminometers are not amenable for customization. The users are able to adapt, expand the potential, and customize the LuminoCell within open-source philosophy with forward compatibility by the scientific community. For example, one can expand the number of sensors to measure more samples simultaneously, or replace the TSL237S-LF with another sensor that would be better suited for the particular experimental system or reporter.

These abilities make the LuminoCell an applicable platform for a wide range of experimental approaches that use luciferase reporter systems. Our work has enhanced the availability of the luminometer system with improved expandability and customizability,

benefiting researchers from a wide range of life sciences disciplines.

## Conclusions

This study provides a full description of affordable, versatile, and open-source platform (LuminoCell) that is capable to monitor luciferase activity in living cell cultures. In addition, we provide proof-of-concept experiments of the device use, in monitoring the activity of cell signalling pathways, cytotoxicity assays, and gene expression assays.

# Materials and Methods

### Cell culture and treatments

All cell lines were cultured in Knockout DMEM (Invitrogen, Life Technologies Ltd.), containing 10% FBS, (PAA), 2 mM L-glutamine (Invitrogen, Life Technologies Ltd.), 1 × MEM non-essential amino acid solution, 1 × penicillin/streptomycin (PAA), and 10 $\mu$M $\beta$-mercaptoethanol (Sigma-Aldrich). The cells were incubated at 37°C/5% CO$_2$ and regularly passaged using trypsin. For cell treatment, the following inhibitors and growth factors were used: FGF (233-FB-025; R&D Systems), EGF (SRP3196-500UG; Sigma-Aldrich), TNF-$\alpha$ (210-TA-005; R&D Systems), CHIR99021 (HY-10182; MedChem Express), and vemurafenib (Hy-12057; MedChem Express). Concentrations are indicated in each experiment.

### Luciferase vectors and reporter cell lines

For in-cell monitoring of the EGF and FGF signalling pathway activity, the 293T::pKrox24$^{Luc}$ and RCS::pKrox24$^{Luc}$ cells were produced from wt 293T and RCS cells using a piggyBac transposase for stable integration of TR01F plasmid (Mossine et al, 2013). NF-$\kappa$B responsive element was replaced by Mapk-ERK responsive element from the pKrox24(MapErk)$^{Luc}$ reporter (Gudernova et al, 2017). For assessment of the NF-$\kappa$B pathway activity, the original TR01F was stably integrated into the 293T cells, again using the piggyBac transposase resulting in the 293T::NF-$\kappa$B$^{Luc}$ cells (Mossine et al, 2013). The clones with a successfully integrated cassette were selected as puromycin resistant and stimulated with appropriate ligands. Clones from each cell line with the best signal/noise ratio were used for the experiments. Plasmid pGL4.50(luc2/CMV/Hygro) (Promega) was used for generation of the A375-IV$^{Luc}$ cell line by antibiotic selection (Hygromycin b, 200 $\mu$g/ml, 31282-04-9; Santa Cruz Biotechnology) and the limiting dilution method for clonal lines generation. Super-Top-Flash 293T (STF) cells were a kind gift from Q Xu and J Nanthas (Johns Hopkins University) (Xu et al, 2004). For *BMAIL1* expression monitoring, hDfs were transduced using lentiviral particles containing the pABpuro-BluF vector (pABpuro-BluF was a kind gift from Steven Brown [plasmid #46824; Addgene; http://n2t.net/addgene:46824; RRID:Addgene_46824]) (Brown et al, 2005). Lentiviral particles were generated, as described in Peskova et al (2019, 2020). Upon transduction, hDf::BMAL1$^{Luc}$ were cultured in the presence of Puromycin (0.7 $\mu$g/ml).

Human retinal epithelia cell line ARPE-19 was transduced using lentiviral particles containing the NanoLuc (Vector builder). ARPE-19 cells were selected using puromycin (1 µg/ml). For measurement, the activity of NanoLuc the culture medium was supplemented with 100x diluted Nano-Glo EndurazineTM Live Cell Substrate (N2571; Promega).

## Measuring luciferase activity

For continuous measurement using the LuminoCell: If not stated otherwise, 200,000 cells/well were seeded into a 40-mm cell culture Petri dish and the cells were allowed to grow for an additional 48 h. Before each experiment, the cell culture medium was replaced with the medium of the same composition but without phenol red containing 100 µM D-Luciferin sodium salt (L6882; Merck). The source code (https://github.com/barta-lab/LuminoCell) was compiled using Arduino IDE software (v. 1.8.13) (www.arduino.cc) and uploaded into Arduino MCU using a USB cable. The luminometer was placed into a tissue incubator, the culture dishes were positioned into the luminometer and covered by a lid to protect the cells from potential incoming light. Luciferase activity was measured using a built-in serial monitor in the Arduino IDE software on a laptop computer Lenovo SL500 with an Ubuntu operating system (20.04 version). The measured values were saved into a CSV file and processed in R studio (version 1.3.1093). For serum shock treatment, the confluent culture of the hDf::BMAL1$^{Luc}$ was kept for 48 h in serum-free medium, then treated with high-serum concentrated medium (50%) for 2 h, washed and replaced with serum-free (phenol free, 100 µM D-Luciferin sodium salt) medium. For hDf::BMAL1$^{Luc}$ data analysis, the data were analyzed using a single cosinor-based method (Cornelissen, 2014) with a constant period of 30 h measured with the BioDare2 (Zielinski et al, 2014). The analysis was carried out in Prism 8 software (GraphPad).

For end-point luciferase activity measurement using a benchtop luminometer: The STF cells were treated with DMSO or 2.5 µM CHIR99021 for 3, 8, and 24 h. The A375-IV$^{Luc}$ cells were exposed to 2.5 µM vemurafenib for 24, 48, and 72 h. After that time, the cells were washed with PBS, lysed, and subsequently an assay was performed accordingly to the manufacturer's protocol (E1960; Promega) and luminescence was detected using the Hidex Bioscan Plate Chameleon Luminometer (Hidex).

## Western blot analysis

The cells were harvested into the sample buffer (125 mM Tris–HCl pH 6.8, 20% glycerol, 4% SDS, 5% β-mercaptoethanol, and 0.02% bromophenol blue). The samples were resolved by SDS–PAGE, transferred onto a polyvinylidene difluoride (PVDF) membrane, incubated with the primary antibodies (see the list of antibodies in Table S2) and with an anti-rabbit secondary antibody (Merck), and visualized by chemiluminescence substrate (Merck) using a Fusion Solo station (Vilber).

## Luminometer description and construction

Files for the 3D print, including the Arduino source code, are located in the GitHub repository (https://github.com/barta-lab/LuminoCell).

### Technical notes

**The sensor** The TSL237S-LF (Mouser; and alternative vendors from Asia) sensor is a temperature compensated sensor which responds over the light range of 320–1,050 nm. The sensor has an acceptable angle of 120° and therefore is positioned in the centre of a well, at a 13 mm distance from the sample (bottom of the cell culture dish) to gather light coming from the whole surface area of a dish (Fig S2). We recommend adding a decoupling capacitor (at least 1 µF) between the ground (GND) and Vdd leading to a reduction of noise and potential voltage spikes.

**Software interrupts and MCU** For pulse counting, we used the interrupt approach. When an external interrupt is triggered (in the case of a pulse generated by the sensor), the MCU will cease the operation of the running code and pass control to an associated interrupt service. Therefore, in the case of bright light coming into the sensor (e.g., a sunny day), the microcontroller may not proceed with the programme and the serial output displays no values. This has some implications for simultaneous reading from multiple sensors (wells). If the MCU receives too many pulses from one sensor, it cannot count pulses from the other sensor(s). However, this is not an issue for luciferase activity measurement, as the highest frequency of incoming pulses that we measured in this study was 0.47 Hz (142 pulses/5 min). Therefore, it is highly unlikely that in the case of luciferase measurement the incoming interrupts will prevent readings from other sensors.

As an MCU any Arduino developmental board can be used. We tested Arduino Nano Every, Arduino Nano, Arduino Micro, and Arduino Uno. However, the choice of MCU largely depends on the number of pins on the MCU capable of performing interrupt operations (for the details check the Arduino website https://www.arduino.cc/reference/en/language/functions/external-interrupts/attachinterrupt/).

**3D design** The LuminoCell consists of three parts that are printed on a conventional 3D printer: (I) lid, (II) middle part, and (III) bottom part (Figs 1C and S2). The lid contains 1 mm openings at its bottom rim that are embedded into a groove upon closing the lid. This prevents incoming light, while allowing gas and moisture exchange. The bottom and middle parts are assembled using four 3 mm self-tapping screws. The gap between the bottom and middle part is then sealed using silicon glue to prevent an increment of moisture inside the device.

**Construction notes** Before you begin to assemble the LuminoCell, remove all indication LEDs from the MCU, as they may produce light that might interfere with the sensors (for the positions of the LEDs, check the MCU manual). Alternatively, place an MCU into a separate box outside a tissue incubator. In this case keep the wire(s) between MCU and sensors as short as possible. Secure the MCU on the bottom part using spacers (or any other preferred method). Connect the sensors with the MCU and capacitor using wires according to the scheme (Fig S1). Connect the MCU with a USB cable. Assemble the bottom and middle part using 3 mm self-tapping screws. Cover the LuminoCell with the lid. Install Arduino IDE software (www.arduino.cc) onto your computer and run it. Connect the LuminoCell with a computer using the USB cable. Download the

source code (.ino file) from https://github.com/barta-lab/LuminoCell and open it in Arduino IDE. Upload the code to the MCU and run the serial monitor (Ctrl + Shift + M). You should see the number of pulses that are detected in each well/sensor. The default integration time is set to 300,000 ms (5 min). If everything works and you plan to use the LuminoCell in a tissue incubator, seal the bottom part and middle part with silicon glue, additionally seal all holes in the wells for the sensor leads and opening for the USB cable.

## Data Availability

The 3D design data and source code from this publication have been deposited to the GitHub repository https://github.com/barta-lab/LuminoCell. Other data are available upon reasonable request.

## Supplementary Information

## Acknowledgements

T Bárta is supported by the Czech Science Foundation (GA21-08182S) the Grant Agency of Masaryk University (GAMU) – MUNI/G/1391/2018. A Hampl is supported by the European Regional Development Fund - Project INBIO (No.CZ.02.1.01/0.0/0.0/16_026/0008451). P Krejčí is supported by the Ministry of Education, Youth and Sports of the Czech Republic (LTAUSA19030), the Agency for Healthcare Research of the Czech Republic (NV18-08-00567), and the Czech Science Foundation (GA19-20123S, 21-26400K). P Macháčková is supported by MEYS CR (LM2018129 Czech-BioImaging). V Bryja is supported by the Czech Science Foundation (GX19-28347X).

### Author Contributions

K Weissová: conceptualization, data curation, formal analysis, validation, investigation, visualization, methodology, and writing—original draft, review, and editing.
B Fafílek: conceptualization, data curation, formal analysis, methodology, and writing—original draft, review, and editing.
T Radaszkiewicz: conceptualization, data curation, formal analysis, visualization, and writing—original draft, review, and editing.
C Celiker: data curation, formal analysis, validation, visualization, and writing—original draft, review, and editing.
P Macháčková: conceptualization, resources, investigation, and writing—original draft, review, and editing.
T Čechovfá: data curation, formal analysis, visualization, and writing—original draft, review, and editing.
J Šebestíiková: data curation, formal analysis, and writing—original draft, review, and editing.
A Hampl: funding acquisition and writing—original draft, review, and editing.
V Bryja: conceptualization, data curation, supervision, funding acquisition, investigation, methodology, and writing—original draft, review, and editing.

P Krejčí: conceptualization, data curation, funding acquisition, investigation, methodology, and writing—original draft, review, and editing.
T Bárta: conceptualization, resources, data curation, software, formal analysis, supervision, funding acquisition, validation, investigation, visualization, methodology, project administration, and writing—original draft, review, and editing.

## Conflict of Interest Statement

The authors declare that they have no conflict of interest.

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
