## [Reviewer comments · Life Science Alliance]

Life Science Alliance

LuminoCell: Versatile and Affordable Platform for Real-Time Monitoring of Luciferase-Based Reporters

Kamila Weissová, Bohumil Fafílek, Tomasz Radaszkiewicz, Canan Celiker, Petra Macháčková, Tamara Čechová, Jana Šebestíková, Ales Hampl, Vitezslav Bryja, Pavel Krejčí, and Tomas Barta

DOI: <https://doi.org/10.26508/lsa.202201421>

Corresponding author(s): Tomas Barta, Masaryk University

Review Timeline:

Submission Date:	2022-02-23
Editorial Decision:	2022-03-31
Revision Received:	2022-04-05
Editorial Decision:	2022-04-07
Revision Received:	2022-04-08
Accepted:	2022-04-08

Scientific Editor: Novella Guidi

Transaction Report:

March 31, 2022

Re: Life Science Alliance manuscript #LSA-2022-01421

Dr. Tomas Barta
Masaryk University
Department of Histology and Embryology
Kamenice 5
Brno 62500
Czech Republic

Dear Dr. Barta,

Thank you for submitting your manuscript entitled "LuminoCell: Versatile and Affordable Platform for Real-Time Monitoring of Luciferase-Based Reporters" to Life Science Alliance. The manuscript was assessed by expert reviewers, whose comments are appended to this letter. We, thus, encourage you to submit a revised version of the manuscript back to LSA that responds to all of the reviewers' points.

Thank you for this interesting contribution to Life Science Alliance. We are looking forward to receiving your revised manuscript.

Sincerely,

B. MANUSCRIPT ORGANIZATION AND FORMATTING:

Reviewer #1 (Comments to the Authors (Required)):

The paper describes the experience with TSL237S-LF light detector in application to the luciferase based reporters in several biological application. The work can be qualified as novel sufficiently for publication in spite of the fact that this kind of detectors has been reported earlier for light measurement application in biology (Droujko J., Molnar P. Open-Source, Low-Cost, In-Situ Turbidity Sensor for River Network Monitoring: preprint. In Review, 2021, Pereira C.B. et al. Construção de um luxímetro digital utilizando plataforma Arduino para uso em laboratórios didáticos // Rev. Bras. Ensino Fís. 2021. Vol. 43. P. e20200502). Authors providing accurate data proving, that suggested device suitable to develop affordable luminometer which allows continuous measurement of luciferase activity in some useful area of sensibility. I would suggest authors to add conclusions section to the paper.

Reviewer #2 (Comments to the Authors (Required)):

The manuscript "LuminoCell: Versatile and Affordable Platform for Real-Time Monitoring of Luciferase-Based Reporters" describes a new real-time luminometer device. The authors provide a construction plan including links for downloadable files (for the 3D print, Arduino microcontroller unit source code) for self-assembly and show that their developed system is highly cost-effective. As this device offers the possibility for a small and cheap real-time reporter system that can be placed directly into cell culture incubators, it is a great opportunity for research under a tight budget. Compared to classic luminometer analysis, that is an end-point analysis, this new device allows real time monitoring of the reporter system thus optimization of timing of end-point is greatly reduced. The authors further show representative applications (major signaling pathways, cytotoxicity assay and circadian oscillations) using different reporter cell lines and vectors, demonstrating the device's versatility. They also compare the data collected with their device to classic end-point luminometer measurements and western blot analysis further supporting the reliability of their device.

The information provided about the assembly of the device (at least from my point of view as a molecular biologist) seem detailed enough to build this device in your own lab with little or no experience. They provide a detailed explanation about the construction with additional notes in the Material and Methods section that is for sure very helpful during assembly.

The authors clearly show the versatility of their device as they tested it for the activation of major signaling pathways (FGFR, EGFR, Wnt, NF- κ B; Figure 2E, 2G, 3A, 3G), for a cytotoxicity assay (Figure 3D) and they also were able to measure circadian oscillations (Figure 4A).

They also address potential difficulties of the measurement (low number of cells, low sensitivity of detector, Figure 2D+4A) but they show that with certain modifications in the measurement parameters these shortcomings can be counteracted.

Overall the data presented in the manuscript strongly support the low price, versatility, accuracy and reliability of their device.

Figure 2B supposedly shows normalized data. It's not stated to what the data was normalized and it is a bit strange that the curve then still goes higher than 100.

Figure 2E should be bigger, the colors are really hard to distinguish in this size

In the discussion they reference Figure 4B when talking about increasing the integration time but 4B is the western blot, therefore I assume it should be Figure 4A.

I assume that the 40mm cell culture dishes can be replaced with new samples after measurement, therefore the assembly of the device only has to be performed once, but it should be stressed more in the manuscript that the device can be reused with new cell samples multiple times.

Dear Editor and Reviewers,

Thank you very much for your comments on our manuscript, entitled "**LuminoCell: Versatile and Affordable Platform for Real-Time Monitoring of Luciferase-Based Reporters**" (Manuscript ID: LSA-2022-01421). All the comments and suggestions were very helpful for improving the quality of the manuscript. All changes to the manuscript are highlighted in red colour.

We have revised the manuscript by following all the comments carefully. Our responses to all the comments, including the details on how the manuscript has been revised, are provided in the following Responses to Reviewers' Comments section.

Reviewers' Comments to Author:

Reviewer: 1

"The work can be qualified as novel sufficiently for publication in spite of the fact that this kind of detectors has been reported earlier for light measurement application in biology (Droujko J., Molnar P. Open-Source, Low-Cost, In-Situ Turbidity Sensor for River Network Monitoring: preprint. In Review, 2021"

Response: Thank you for pointing us to this work, we were not aware of this study. We added a reference of this work (see page 2, line 35).

„I would suggest authors to add conclusions section to the paper.“

Response: We added a section "Conclusion" (see page 6, line 11), additionally we added "Summary Blurb", as required by LSA journal (see page 1, line 36).

Reviewer: 2

"Figure 2B supposedly shows normalized data. It's not stated to what the data was normalized and it is a bit strange that the curve then still goes higher than 100."

Response: The data is normalized to the number pulses detected before the addition of FGF2 (15 pulses/5 minutes before normalization => 0 pulses/5 minutes after normalization), the maximal number of pulses detected was 142 pulses/ 5 minutes before normalization => 127 pulses/5 minutes after normalization. We added a sentence that should clarify the background subtraction/normalization (see page 3, line 22).

"Figure 2E should be bigger, the colors are really hard to distinguish in this size."

Response: We updated the Figure 2E.

"In the discussion they reference Figure 4B when talking about increasing the integration time but 4B is the western blot, therefore I assume it should be Figure 4A."

Response: This escaped to our attention, thank you for the comment. We corrected the reference.

"I assume that the 40mm cell culture dishes can be replaced with new samples after measurement, therefore the assembly of the device only has to be performed once, but it should be stressed more in the manuscript that the device can be reused with new cell samples multiple times."

Response: We do agree that this should be highlighted in the manuscript. We added two sentences (see page 2, line 46).

Please note that in order to meet the requirements raised by LSA journal (uncropped RAW blot images) we updated Figures S3, S4, S5, and S6.

We thank the editor and all reviewers for your comments, which have greatly helped us improve the quality of the paper.

Yours sincerely,

Tomas Barta & co-authors

April 7, 2022

RE: Life Science Alliance Manuscript #LSA-2022-01421R

Dr. Tomas Barta
Masaryk University
Department of Histology and Embryology
Kamenice 5
Brno 62500
Czech Republic

Dear Dr. Barta,

Thank you for submitting your revised manuscript entitled "LuminoCell: Versatile and Affordable Platform for Real-Time Monitoring of Luciferase-Based Reporters". We would be happy to publish your paper in Life Science Alliance pending final revisions necessary to meet our formatting guidelines.

- please make sure that the author order in our system matches the author order in the manuscript
- please consult our manuscript preparation guidelines <https://www.life-science-alliance.org/manuscript-prep> and make sure your manuscript sections are in the correct order
- please use the [10 author names, et al.] format in your references (i.e. limit the author names to the first 10)
- please upload your tables in editable doc or excel format or include them at the end of your manuscript doc file
- please add a callout for table s2 in your main manuscript text

A. FINAL FILES:

B. MANUSCRIPT ORGANIZATION AND FORMATTING:

Sincerely,

April 8, 2022

RE: Life Science Alliance Manuscript #LSA-2022-01421RR

Dr. Tomas Barta
Masaryk University
Department of Histology and Embryology
Kamenice 5
Brno 62500
Czech Republic

Dear Dr. Barta,

Thank you for submitting your Resource entitled "LuminoCell: Versatile and Affordable Platform for Real-Time Monitoring of Luciferase-Based Reporters". It is a pleasure to let you know that your manuscript is now accepted for publication in Life Science Alliance. Congratulations on this interesting work.

DISTRIBUTION OF MATERIALS:

Again, congratulations on a very nice paper. I hope you found the review process to be constructive and are pleased with how the manuscript was handled editorially. We look forward to future exciting submissions from your lab.

Sincerely,
